# Effects of Meteorological Factors on Apple Yield Based on Multilinear Regression Analysis: A Case Study of Yantai Area, China

**Xirui Han** , **Longbo Chang, Nan Wang, Weifu Kong and Chengguo Wang** *

Yantai Institute, China Agricultural University, Yantai 264670, China
* Correspondence: wangcgyt@cau.edu.cn

**Abstract:** Evaluating the impact of different meteorological conditions on apple yield and predicting the future yield in Yantai City is essential for production. Furthermore, it provides a scientific basis for the increase in apple yield. In this study, first, a grey relational analysis (GRA) was used to determine the quantitative relationship between different meteorological factors and meteorological yield which is defined as affected only by meteorological conditions. Then, the comprehensive meteorological factors extracted by a principal component analysis (PCA) were used as inputs for multiple linear regression (MLR). The apple yield accuracy was compared with the lasso regression prediction. Trend analysis showed that the actual apple yield increased annually, but the meteorological yield decreased annually over a long time. Correlation ranking illustrated that the meteorological yield was significantly correlated with the frost-free period, the annual mean temperature, the accumulated temperature above 10 °C, etc. The good consistency between GRA and MLR–PCA showed that the accumulated temperature above 10 °C, the March–October mean temperature, and the June–August mean temperature are key meteorological factors. In addition, it was found that the principal components $F_2$, $F_4$, and $F_5$ were negatively correlated with meteorological yield, while the principal components $F_1$ and $F_3$ were positively correlated with meteorological yield. Moreover, the MLR–PCA model predicted the apple yield in 2020 as 47.256 t·ha$^{-1}$ with a 7.089% relative error. This work demonstrates that the principal component regression model can effectively extract information about different meteorological factors and improve the model's accuracy for analyzing key meteorological factors and predicting apple yield.

**Keywords:** Yantai area; principal component analysis; multiple linear regression; grey relational analysis; meteorological factors; apple yield



## 1. Introduction

Climate change affects fruit production in some regions in China and worldwide by changing agricultural climate conditions [1], directly affecting fruit yields and quality [2]. Meanwhile, China is the largest apple producer in the world with its northern regions contributing more than 70 percent of the country's apple yield [3,4]. Seasonal climate change is evident and has led to limitations in apple yield in northern China [5]. Some studies have indicated that climate change, including global warming and frost disasters, might induce tremendous decreases in fruit yield. A recent report by Benlloch-González et al. [6] showed that every 4 °C increased in ambient temperature resulted in a reduction in fruit setting and a significant decrease in fruit yield. Zhu et al. [7] discovered that the Loess Plateau area had experienced frost disasters during the apple flowering period, leading to a significant decline and even extinction of apple yield. Cui et al. [8] employed different El Niño southern oscillation patterns to reveal La Niña years promoted apple yield, while El Niño years inhibited apple yield. Many studies have shown that meteorological factors such as accumulated temperature, precipitation, and frost can affect the growth and yield of apples [9,10]. Sen et al. [11] found that minimum temperatures in January,

February, and November, rainfall in December, and maximum temperatures in March and October were important factors affecting apple yield. However, studies have shown that appropriate measures can increase fruit production under trends of future climate change [12–14], especially at middle and high latitudes [15,16]. Apple yield is affected by a variety of meteorological factors. Therefore, identifying the varied characteristics of the different meteorological factors and their impact on apple yield is of great significance for the scientific development of agricultural management measures to ensure stable and increased apple yield.

At present, scholars in different countries have carried out relevant studies on the influence of climate change on apple yield and quality. However, most of the research focuses on the influence of meteorological factors on apple yield in a certain phenological period, ignoring the influence of meteorological factors throughout the whole growing period of apples. Unterberger et al. [17] applied the combination of the phenological series model and Austrian climate prediction to show that spring blocking would seriously affect apples' flowering and fruit setting rates. Delgado et al. [18] researched the effect of climate change on apple phenology in northwest Spain and demonstrated that climate change could affect the start date of internal dormancy and offset the advancing effect of phenology. Funes et al. [19] manipulated the dynamic model to study the meteorological factors affecting the flowering period in the downstream of the Fluvia watershed, revealing the cold and heat demand of crops during the flowering period and improving the credibility of the prediction of the flowering period. Che et al. [20] studied the effect of climate change on apple yield in each phenological period in the Longdong region. Jing et al. [21] applied an HP filter combined with a BP neural network to study the effect of meteorological factors on early apple yield in Yuncheng City. The overall increase in research has led to significant achievements in research outcomes. However, most of the previous studies have used meteorological data directly and there is a need for more analysis of the degree of coverage between data. Qu et al. [22] used grey relational analysis to assess the relationship between apple quality and climatic factors in different apple-producing areas in Shaanxi. Li et al. [10] identified the relationship between climate factors, circulation indicators, and apple yield in China's major apple-producing areas. Demestihas et al. [23] developed a simulation model to study the collaborative services among multiple ecosystems in apple orchards that provide a sustainable system. Yao et al. [24] utilized a sigmoid curve to simulate the relationship between apple quality and yield and meteorological factors.

Currently, simulation models, including the planting density model, crop spatial distribution model, remote sensing simulation model, and evapotranspiration model, are widely applied in the study of crop spatial responses to climate change [25–28]. However, the relevant models depend highly on data acquisition precision, resolution, experimental area, model parameter adjustment, and their application scope. Therefore, these model algorithms need to be selected according to different regions, and at the same time, the model's parameters need to be constantly adjusted [29–31]. Due to the limitations of the above common models, appropriate indicators should be selected to quantify the climate resources that have eliminated soil and fertilization effects. Additionally, indicators are based on expert opinions and literature reviews in agricultural meteorology and ecology, including temperature, precipitation, accumulated temperature, and humidity. For example, the GLAM model was employed by Challinor et al. to quantify the impact of high-temperature threshold exceedance on crops [32]. Based on factors of temperature, precipitation, frost, and soil humidity, López-Morales et al. [33] presented an Internet of Things (IoT) architecture, and usage of this architecture to improve decision making in the agricultural industry using meteorological information. Furthermore, Rosbakh et al. [34] applied climate data to study how Siberian plants shifted their phenology under climate change. This method is easy to apply in terms of expert evaluation and literature review. However, due to the differences in study areas and the subjectivity of index selection, it is necessary to introduce statistical methods and detect the coverage degree of indicators, making the indicators' selection more scientific and reliable.

As the main apple-producing area in China, the apple yield in Shandong Province accounted for 21.6% of the total apple yield in 2020 [35]. However, the spatial distribution of Shandong apple production is uneven. The main production areas are mostly concentrated in the eastern coastal areas of Shandong Province, led by Yantai city, which was rated as the third batch of agricultural characteristics of the advantageous areas [36].At the same time, the Yantai apple has been awarded the title of "the best brand in China's fruit industry" for thirteen consecutive years [37]. Therefore, the sustainable and healthy development of the Yantai apple industry plays an essential role in leading the progress of the Shandong apple industry. In this study, it is hypothesized that:

(i)    For non-stationary yield data, they are thought to be near stationary over an appropriate small interval (5 years is chosen as a small interval in this paper);
(ii)   The collinearity of different meteorological factors requires dimensionality reduction;
(iii)  There is a multivariate linear relationship between the separated obtained meteorological yields and the meteorological factors.

This work aims to investigate the effects of different meteorological factors on apple yield by analyzing meteorological and yield data for thirteen consecutive years in Yantai, Shandong Province. On the basis of clarifying the changing trend of the meteorological yield of apples, the multiple linear regression based on principal component analysis (MLR–PCA) method is used to eliminate the problem of multicollinearity. This work hopes to explore the key meteorological factors affecting meteorological yield and forecast apple yield. The aim of this study is to provide a scientific basis for coping with climate change, improving agricultural production and management patterns, and ensuring increased apple yields.

## 2. Materials and Methods

### 2.1. Study Area

Yantai is located in the northeast of Shandong Province, bordering the Bohai Sea and the Yellow Sea, with a typical temperate continental monsoon climate. The terrain is mostly low and hilly [38], with the hilly area accounting for 76.32% of the total area (Figure 1). The mean annual temperature of the study area is 12.9 °C, the annual precipitation is 661 mm, the frost-free period is 227.7 days, the accumulated temperature above 10 °C is 4567.4 °C/$a$, and the relative humidity in July is 75.7%. The favorable climate and geographical environment in this area has created the famous "Apple Capital of China" [39].

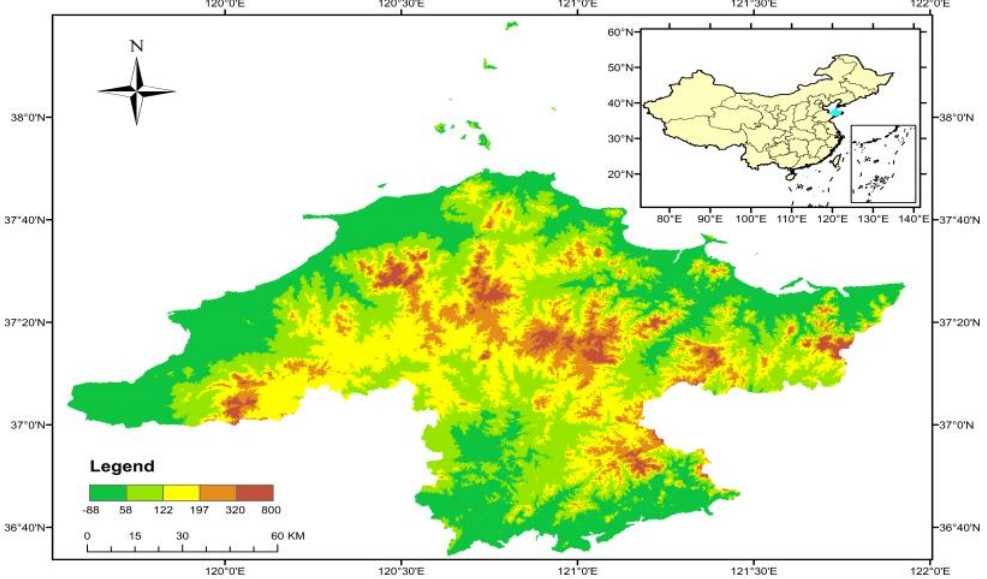

**Figure 1.** A map showing the location of Yantai city in Shandong Province, China (coordinates: 119°34′–121°57′ E longitude and 36°16′–38°23′ N latitude).

*2.2. Data*

Long-term monthly climate data, gathered from eight meteorological stations in Yantai City, Shandong Province, China, from 1999 to 2019, were obtained from the Shandong Bureau of Statistics [40]. These data were organized in a database containing twelve indicators (Table 1) obtained from existing research experience and local conditions [5,41,42]. Statistics including corn yield and acreage data, from 1999 to 2019, were obtained from the Shandong Bureau of Statistics.

**Table 1.** The description of the meteorological indicators employed in this study.

| Variable Name | Unit | Notation |
|---|---|---|
| Annual mean temperature | °C | AAT |
| The annual difference in temperature | °C | AR |
| March–October mean temperature | °C | MATTT |
| The coldest month mean temperature | °C | LAMT |
| Annual average precipitation | mm | MAP |
| The frost-free period | d | FFP |
| The accumulated temperature above 10 °C | °C/a | ATA10 |
| Mid-January mean temperature | °C | ATMJ |
| Annual extreme low temperature | °C | AMT |
| June–August mean temperature | °C | MATSE |
| June–August precipitation | mm | MTPSE |
| July relative humidity | % | RHJ |

Since the statistical yearbook of the Shandong Province only gives the municipal-level yield, it is not realistic to study the eight meteorological stations separately. Therefore, we adopted the equal-weighted average method [43] to convert the original data of eight meteorological stations into meteorological data.

*2.3. Research Methods*

2.3.1. Yield Separation Model

Social and natural factors mainly influence the apple yield. In the long-term production time series of apples, the influence of social factors is reflected in the improvement of production capacity caused by the strengthening of science and technology. The fluctuation of apple yield caused by the improvement of productivity levels is called the trend yield. The influence of natural factors is mainly manifested in the variation of apple yield caused by the difference in interannual meteorological conditions. The fluctuation of apple yield caused by changes in the meteorological conditions is called meteorological yield. Meanwhile, the variation of apple yield that is caused by other factors is called random yield. Therefore, apple yield can be further classified into trend yield, meteorological yield, and error yield [9]. The following equation can explain yield separation:

$$y = y_t + y_c + \varepsilon \qquad (1)$$

where $y$ is the actual yield (t·ha$^{-1}$), $y_t$ is trend yield (t·ha$^{-1}$), $y_c$ is the meteorological yield (t·ha$^{-1}$), and $\varepsilon$ is the error yield (t·ha$^{-1}$) which is usually negligible.

To separate the trend yield sequence from the actual yield, a five-year moving average method, also known as 5a sliding average, was applied to fit the trend yield [44,45]. The calculation of 5a sliding average is shown by the following equation:

$$y_i = a_i t + b_i \ (i = n - K + 1) \qquad (2)$$

where $i$ denotes the number of equations, $K$ denotes the moving step length, $n$ denotes the number of years, and $t$ denotes time.

The moving average included the four preceding years and the year of interest, and was calculated as follows:

$$\hat{y}_t(t) = \frac{1}{5}\sum_{i-4}^{i} y_i(t) \tag{3}$$

where $y_i(t)$ denotes the value in the year of interest and $\hat{y}_t(t)$ denotes the five-year moving average of the year of interest.

### 2.3.2. Screening Model for Key Meteorological Factors

The meteorological yield of apples was taken as the parent sequence $\{x_0(k)\}$. Each meteorological factor is taken as the sub-sequence $\{x_i(k)\}$. $k$ represents the number of samples. Calculating the correlation coefficient of $\{x_i(k)\}$ and $\{x_0(k)\}$ as:

$$\xi_i(k) = \frac{\min_i\min_k \Delta_i(k) + \rho\max_i\max_k \Delta_i(k)}{\Delta_i(k) + \rho\max_i\max_k \Delta_i(k)} \tag{4}$$

where $\Delta_i(k) = |x_i(k) - x_0(k)|$ is the absolute difference between $\{x_i(k)\}$ and $\{x_0(k)\}$ at the $k$-th term at $i$-th point. $\rho$ represents resolution factor, generally $\rho = 0.5$.

### 2.3.3. Quantifying the Relationship between Meteorological Factors and Yield

Multiple linear regression based on the principal component analysis model (MLR–PCA) uses indicators obtained from a PCA as the inputs. A PCA adopts the idea of dimensionality reduction to simplify multiple indicators into a few representative comprehensive indicators that can reflect most of the information, so it avoids the possible multicollinearity problem among multiple indicators [46,47].

(1) To eliminate the effect of dimension of different meteorological factors and apple yield by Z-Score standardization. The following equation can explain the standardized calculation:

$$D_i = \frac{x_i - \overline{x}_i}{S_i} \tag{5}$$

where $\overline{x}_i$ is the mean value of the $i$-th data set, $S_i$ is the standard deviation of the $i$-th data set.

(2) Correlation analysis is a method used to calculate the degree of correlation between two variables. The correlation between each meteorological factor and yield and each meteorological factor was determined by correlation analysis. The correlation coefficient $R$ was defined as:

$$R = \frac{n\sum_{i=1}^{n} x_i y_i - \sum_{i=1}^{n} x_i \sum_{i=1}^{n} y_i}{\sqrt{n\sum_{i=1}^{n} x_i^2 - \sum_{i=1}^{n}(x_i)^2} \cdot \sqrt{n\sum_{i=1}^{n} y_i^2 - \sum_{i=1}^{n}(y_i)^2}} \tag{6}$$

where $n$ is the number of meteorological factor and yield, and $x_i$, $y_i$ are two sets of data to be judged.

(3) Multicollinearity refers to a linear regression model in which the estimation of the model is distorted or difficult to estimate accurately due to the high correlation between the explanatory variables. To avoid model distortion, variance inflation factor analysis ($VIF$) is applied to test for multicollinearity through the equation:

$$VIF_m = \frac{1}{1 - R_{1\sim k\backslash m}^2} \tag{7}$$

where the maximum $VIF_m$ value in excess of 10 is often taken as an indication that multicollinearity may be unduly influencing the least square estimate.

(4)  KMO test statistic and Bartlett spherical test are adopted to judge the correlation between indicators and determine whether the variables are suitable for a principal component analysis. Assuming that there are $n$ groups of samples and $P$ indicators, a standard matrix $V$ of size $N \times P$ can be formed. Then, the first, second, ... $i$-th ($i < n$) principal components corresponding to the eigenvalues with a cumulative contribution of more than 85% are generally taken. The expression of the principal component is calculated through Equation (8).

$$\begin{bmatrix} F_1 \\ F_2 \\ \vdots \\ F_i \end{bmatrix} = \begin{bmatrix} c_{11} & c_{12} & & c_{1n} \\ & & \cdots & \\ c_{21} & c_{22} & & c_{2n} \\ & \vdots & \ddots & \vdots \\ c_{i1} & c_{i2} & \cdots & c_{in} \end{bmatrix} \begin{bmatrix} v_1 \\ v_2 \\ \vdots \\ v_n \end{bmatrix} \tag{8}$$

where $F_i$ is the $i$-th PC of variables, $c_{in}$ is the loading coefficient that indicates how much the $n$-th variables participate in defining $F_i$, and $v_n$ is the standardized variable.

(5)  Let $y$ be the dependent variable and $F_1$, $F_2...F_k$ be the principal component of $n$ independent variables, then the expression is calculated through equation:

$$y = \beta_0 + \beta_1 F_1 + \beta_2 F_2 + \cdots + \beta_n F_n \tag{9}$$

where $\beta_n$ represents the regression coefficient.

2.3.4. The Trend Yield Forecast Model

Harmonic weight prediction is an algorithm for trend yield extension [48–50]. The specific steps are as follows:

$$w_{t+1} = y_{c,t+1} - y_{c,t} \tag{10}$$

where, $y_{c,t}$ is the trend yield of the $t$-th year. $y_{c,t+1}$ is the trend yield of next year. $w_{t+1}$ is the annual increase of the trend yield.

$$w = \sum_{t=1}^{n-1} c_{t+1} \cdot w_{t+1} \tag{11}$$

where, $c_{t+1}$ is the harmonic weight coefficient, and $c_{t+1}$ is calculated according to the following Equations (12) and (13):

$$c_{t+1} = \frac{m_{t+1}}{n-1} (t = 1, 2, 3, \cdots, n-1) \tag{12}$$

$$m_{t+1} = m_t + \frac{1}{n-t} (m_1 = 0) \tag{13}$$

Then, the prediction of trend yield is calculated by Equation (14).

$$\hat{y}_{c,n+1} = y_{c,n} + w \tag{14}$$

2.3.5. Comparison of Prediction Accuracy

To verify the reliability of the model constructed in this paper, the meteorological output in 2021 was predicted and compared with the results predicted by the lasso regression model [51]. Lasso regression avoids the distortion of the regression results caused by multicollinearity and endogeneity between variables in the regression process by rapidly compressing the coefficients of nonimportant explanatory variables to zero. The objective function of lasso regression can be expressed as:

$$L(b) = \sum (y - Xb)^2 + \eta \|b\|_1 = \sum (y - Xb)^2 + \sum \eta |b| \tag{15}$$

where $\eta\|b\|_1$ is the penalty term of the function, $\eta$ is the penalty coefficient, and $\|b\|_1$ is the regularization of regression coefficient $b$ and represents the sum of the absolute values of all regression coefficients.

## 3. Results and Analysis

### 3.1. Meteorological Yield Separation

The 5a sliding average method was selected to calculate the trend yield [7,43,44]. The trend yield of apples from 1999 to 2019 was counted. Then, the meteorological yield was separated from the actual yield. The results are shown in Figure 2.

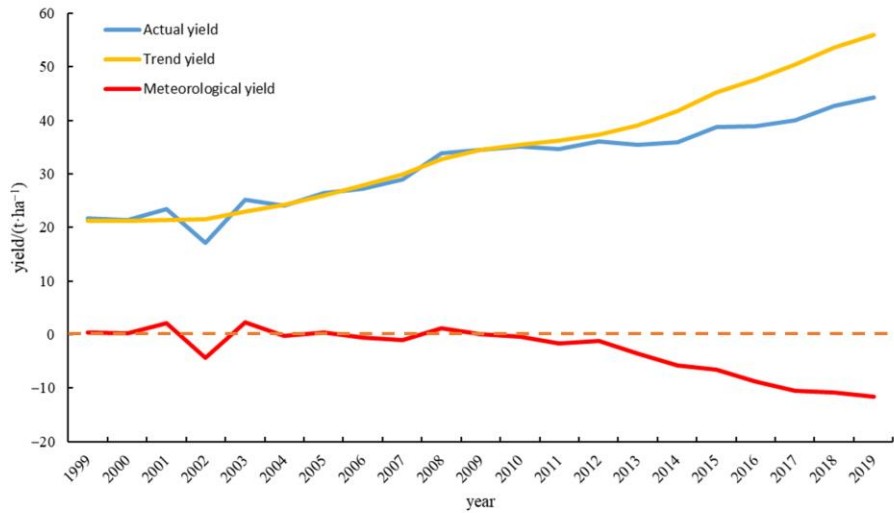

**Figure 2.** Actual, trend and meteorological yield of Yantai apples from 1999 to 2019 (the orange line in the figure where the meteorological yield is equal to 0 t·ha$^{-1}$ is the dividing line where the meteorological yield plays a facilitating or inhibiting role in the actual yield. If it is higher than the orange line, it plays a facilitating role in the actual yield; conversely, if it is lower than the orange line it plays an inhibiting role in the actual yield.

As shown in Figure 2, the actual yield of apples demonstrates a steady upward trend, in particular, since 2014. The apple yield shows an increasing annual trend, which may be related to the development and progress of field management measures, seed selection and seedling raising methods, etc. The meteorological yield shows a declining annual trend, which may be related to meteorological factors caused by severe global climate change and the increasing greenhouse effect in recent years. Meanwhile, as shown in Figure 2, in the early years (before 2014), the meteorological yield was stable and near the 0 t· ha$^{-1}$ line. While in the years after 2014, the meteorological yield was significantly less than 0 t· ha$^{-1}$, which would seriously hinder the growth of apple yield. These data suggest that orchard managers have a low utilization rate of local climate resources and may not consider climate change when making apple production plans.

### 3.2. Screening of Key Meteorological Factors

The KMO statistic test is required before a principal component analysis is used, and the KMO statistic was $0.422 < 0.600$, which is not suitable for variable importance ranking using a principal component analysis [52]. In previous studies by Li et al. [10] and Shen et al. [53], GRA was utilized to screen key meteorological factors. The data of Yantai City from 1999 to 2019 were standardized, and the GRA was used to calculate each index's correlation coefficients in the sub-sequence and the parent sequence. Combined with Equation (4), the average value of the correlation coefficient of each meteorological

factor was taken as the grey relational coefficient between each meteorological factor and the meteorological yield.

$$r_i = \frac{1}{n} \sum_{k=1}^{n} \xi_i(k) \; (k = 1, 2, 3, \cdots n) \tag{16}$$

GRA was conducted on twelve meteorological factors and meteorological yield to obtain the grey correlation coefficient. The results are shown in Figure 3.

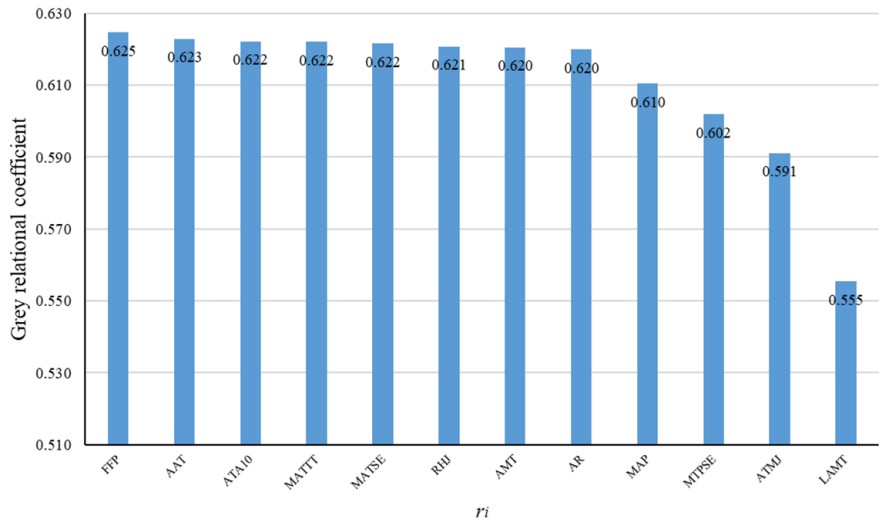

**Figure 3.** The grey correlation coefficient between twelve meteorological factors and yield (variables in the figure present twelve meteorological factors after standardized treatment).

The correlation degree of different meteorological factors was sorted, and the ranking order represents the impact degree of different meteorological factors on apple meteorological yield. Figure 3 illustrates that the twelve meteorological factors affecting apple meteorological yield were sorted as follows: the frost-free period, the annual mean temperature, the accumulated temperature above 10 °C, the March–October mean temperature, the June–August mean temperature, the July relative humidity, the annual extreme low temperature, the annual difference in temperature, the annual average precipitation, the June–August precipitation, the mid-January mean temperature, and the coldest month mean temperature.

Grey correlation analysis can be employed to obtain the correlation ranking among variables, but it is not easy to judge the promoting or inhibiting relationship between variables. Therefore, key factors are initially screened by a grey relational analysis. Generally, two to four variables are selected as key variables [54,55]. The low threshold is set by selecting eight variables as candidates for key meteorological factors based on the initial screening. Thus, the meteorological factors with a correlation coefficient greater than 0.62 are defined as the key meteorological factors. The preliminary screening results include the frost-free period, the annual mean temperature, the accumulated temperature above 10 °C, the March–October mean temperature, the June–August mean temperature, the July relative humidity, the annual extreme low temperature, and the annual difference in temperature. Producers should pay more attention to the changes in the key meteorological factors, timely adjust the field's microclimate, and formulate a reasonable fruit-tree management plan to prevent the dramatic decline of apple yield.

### 3.3. Construction of Meteorological Yield Forecast Model

SPSS 28.0 was applied for the collinearity test, principal component analysis, and multiple linear regression. The correlation and scatter diagrams were conducted with R

version 4.2.2 using the packages dplyr, tidyverse, and ggplot2 for visualization of data in plots and heatmaps [56,57]. The results of the correlation analysis are shown in Figure 4.

**Figure 4.** Correlation coefficient matrix heatmap and significance analysis of twelve meteorological factors and yield (*: $p < 0.05$ **: $p < 0.01$ ***: $p < 0.001$).

Figure 4 demonstrates that, under the two-tailed test conditions, the meteorological yield was significantly correlated with the frost-free period and the accumulated temperature above 10 °C. Additionally, the accumulated temperature above 10 °C had an extremely significant inhibitory effect on the meteorological yield of apples ($p < 0.01$), and the frost-free period had a significant inhibitory effect on the meteorological yield of apples ($p < 0.05$). There were certain correlations among different meteorological factors. For example, the annual mean temperature was extremely significantly positively correlated with the March–October mean temperature, the annual extreme low temperature, and the coldest month mean temperature ($p < 0.01$), while the annual mean temperature was extremely significantly negatively correlated with the annual difference in temperature and the annual average precipitation ($p < 0.01$). The annual difference in temperature was extremely significantly negatively correlated with the coldest month mean temperature, the mid-January mean temperature, the annual mean temperature, and the annual extreme low temperature ($p < 0.01$), while the annual difference in temperature was significantly positive correlated with the annual average precipitation, the June–August precipitation, and the June–August mean temperature ($p < 0.05$). The coldest month mean temperature was extremely significantly positively correlated with the mid-January mean temperature, the annual mean temperature, and the annual extreme low temperature ($p < 0.001$). The correlation analysis shows that the correlation between meteorological factors was strong, but the correlation between meteorological factors and meteorological yield was not high. The multiple regression equation and variance analysis are shown in Equation (17) and Table 2.

$$Y_c = 0.58AAT - 4.47AR - 2.16MATTT - 4.48LAMT - 0.01MAP - 0.03FFP - 0.01ATA10 - 0.34ATMJ$$
$$+0.54AMT + 4.09MATSE + 0.05MTPSE + 0.22RHJ + 86.12 \tag{17}$$

**Table 2.** Analysis of variance between meteorological yield and 12 meteorological factors.

| Analysis of Variance | df | SS | MS | F | Significance F |
|---|---|---|---|---|---|
| Regression Analysis | 12 | 298.225 | 24.852 | 2.106 | 0.148 |
| Residual | 8 | 94.388 | 11.799 | | |
| Total | 20 | 392.613 | | | |

It can be remarked from the results of variance analysis in Table 2 that Significance F value is greater than 0.05, which indicates that the multiple regression equation was not significant and could not be used for prediction. This phenomenon is probably caused by the serious collinearity between meteorological factors. Therefore, the collinearity diagnosis of meteorological factors needs to be checked. The collinearity diagnosis result is shown in Table 3.

**Table 3.** Diagnosis of covariance between different meteorological factors.

| Variables | *VIF* | Tolerance | Variables | *VIF* | Tolerance |
|---|---|---|---|---|---|
| AAT | 9.011 | 0.111 | ATA10 | 7.827 | 0.128 |
| AR | 109.230 | 0.009 | ATMJ | 3.360 | 0.298 |
| MATTT | 10.315 | 0.097 | AMT | 15.831 | 0.063 |
| LAMT | 46.532 | 0.021 | MATSE | 40.978 | 0.024 |
| MAP | 21.330 | 0.047 | MTPSE | 29.226 | 0.034 |
| FFP | 2.975 | 0.336 | RHJ | 4.993 | 0.200 |

Table 3 shows the *VIF* values of different meteorological factors were greater than 10, and the *VIF* values of the annual difference in temperature indicators were far more than 100, indicating that there is serious multicollinearity among different meteorological factors in Yantai. The meteorological factors in the meteorological system will interact and influence each other, so that one meteorological factor can be calculated from other meteorological factors. For example, the annual difference in temperature is obtained by subtracting the mean temperature of the coldest month from the mean temperature of the hottest month in a year. Interaction may be the reason for the serious collinearity of meteorological factors in Yantai City. The direct establishment of regression models will cause serious information coverage problems making the original data unable to be fully utilized. Because of the serious collinearity between meteorological factors in Yantai City, the established multiple regression model is invalid. A principal component analysis reduces the dimensionality of multivariate factors, extracts most of the information of the original data, and reduces the coverage between the information. Therefore, a principal component analysis can solve the problem.

The principal component analysis model was established, and the Bartlett sphericity test was performed on the standardized correlation coefficient matrix. The significance was 0.00 < 0.05; thus, the original hypothesis of Bartlett's sphericity test was rejected, and the principal component analysis could be performed. The meteorological yield and twelve meteorological factors were analyzed using a principal component analysis. The results are shown in Table 4.

Table 4 demonstrates that the cumulative contribution of the first five principal components reached 90.076%, which meets the requirement of a cumulative contribution of >85% for a principal component analysis. It indicates that most of the information in the original data has been extracted at this point. For its principal components, only five principal components were needed to represent the twelve indicators, and the contribution of each variable to the principal components constitutes the principal component matrix, as shown in Table 5.

As shown in Table 5, the principal component $F_1$ is composed of the annual mean temperature, the annual difference in temperature, the coldest month mean temperature, and the annual extreme low temperature. The principal component $F_2$ is composed of the

June–August mean temperature, the March–October mean temperature, and the accumulated temperature above 10 °C. The principal component $F_3$ is composed of the July relative humidity. The principal component $F_4$ is composed of the annual average precipitation and the June–August precipitation. The principal component $F_5$ is composed of the mid-January mean temperature. The principal components $F_1$–$F_5$ can be named as temperature factor, growing season heat factor, humidity factor, precipitation factor, and cooling factor, respectively. The principal component $F_1$–$F_5$ can be expressed as:

**Table 4.** The sum of squared loadings and eigenvalues extracted by the principal component analysis.

| $F_i$ | Eigenvalue | | Extraction Sums of Squared Loadings | | Rotation Sums of Squared Loadings | | |
|---|---|---|---|---|---|---|---|
| | Eigenvalue | Variance Contribution/% | Eigenvalue | Variance Contribution/% | Eigenvalue | Variance Contribution/% | Cumulative Variance Contribution/% |
| 1 | 5.299 | 44.156 | 5.299 | 44.156 | 4.255 | 35.456 | 35.456 |
| 2 | 2.424 | 20.197 | 2.424 | 20.197 | 2.337 | 19.474 | 54.929 |
| 3 | 1.469 | 12.241 | 1.469 | 12.241 | 1.951 | 16.256 | 71.185 |
| 4 | 1.035 | 8.626 | 1.035 | 8.626 | 1.608 | 13.403 | 84.587 |
| 5 | 0.583 | 4.856 | 0.583 | 4.856 | 0.659 | 5.488 | 90.076 |
| 6 | 0.471 | 3.924 | | | | | |
| 7 | 0.353 | 2.944 | | | | | |
| 8 | 0.185 | 1.542 | | | | | |
| 9 | 0.085 | 0.711 | | | | | |
| 10 | 0.057 | 0.472 | | | | | |
| 11 | 0.035 | 0.295 | | | | | |
| 12 | 0.004 | 0.036 | | | | | |

**Table 5.** Load matrix of different meteorological factors.

| Variables | Component | | | | |
|---|---|---|---|---|---|
| | $F_1$ | $F_2$ | $F_3$ | $F_4$ | $F_5$ |
| ZAAT | 0.804 | 0.381 | 0.208 | 0.219 | −0.175 |
| ZAR | −0.818 | 0.45 | 0.023 | −0.28 | 0.003 |
| ZMATTT | 0.48 | 0.776 | 0.21 | 0.169 | −0.099 |
| ZLAMT | 0.918 | −0.163 | 0.104 | 0.254 | 0.123 |
| ZMAP | −0.8 | −0.09 | 0.01 | 0.538 | 0.081 |
| ZFFP | 0.638 | 0.304 | −0.447 | −0.199 | 0.189 |
| ZATA10 | −0.001 | 0.69 | −0.561 | 0.246 | 0.246 |
| ZATMJ | 0.684 | −0.279 | 0.274 | 0.018 | 0.555 |
| ZAMT | 0.799 | −0.135 | 0.132 | 0.284 | −0.28 |
| ZMATSE | −0.06 | 0.867 | 0.3 | −0.071 | 0.037 |
| ZMTPSE | −0.786 | 0.027 | 0.034 | 0.571 | 0.089 |
| ZRHJ | −0.367 | 0.149 | 0.82 | −0.12 | 0.168 |

$$
\begin{cases}
\begin{aligned}
F_1 = & \frac{1}{\sqrt{4.255}}(0.804ZAAT - 0.818ZAR + 0.48ZMATTT + 0.918ZLAMT - 0.8ZMAP + 0.638ZFFP \\
& -0.001ZATA10 + 0.684ZATMJ + 0.799ZAMT - 0.06ZMATSE - 0.786ZMTPSE - 0.367ZRHJ) \\
F_2 = & \frac{1}{\sqrt{2.337}}(0.381ZAAT + 0.45ZAR + 0.776ZMATTT - 0.163ZLAMT - 0.09ZMAP + 0.304ZFFP \\
& +0.69ZATA10 - 0.279ZATMJ - 0.135ZAMT + 0.867ZMATSE + 0.027ZMTPSE + 0.149ZRHJ) \\
F_3 = & \frac{1}{\sqrt{1.951}}(0.208ZAAT + 0.023ZAR + 0.21ZMATTT + 0.104ZLAMT + 0.01ZMAP - 0.447ZFFP \\
& -0.561ZATA10 + 0.274ZATMJ + 0.132ZAMT + 0.3ZMATSE + 0.034ZMTPSE + 0.82ZRHJ) \\
F_4 = & \frac{1}{\sqrt{1.608}}(0.219ZAAT - 0.28ZAR + 0.169ZMATTT + 0.254ZLAMT + 0.538ZMAP - 0.199ZFFP \\
& +0.246ZATA10 + 0.018ZATMJ + 0.284ZAMT - 0.071ZMATSE + 0.571ZMTPSE - 0.12ZRHJ) \\
F_5 = & \frac{1}{\sqrt{0.659}}(-0.175ZAAT + 0.003ZAR - 0.099ZMATTT + 0.123ZLAMT + 0.081ZMAP + 0.189ZFFP \\
& +0.246ZATA10 + 0.555ZATMJ + -0.28ZAMT - 0.037ZMATSE + 0.089ZMTPSE + 0.168ZRHJ)
\end{aligned}
\end{cases}
\tag{18}
$$

The principal components $F_1$–$F_5$ were taken as independent variables, and the method of multiple linear regression was adopted to perform an ordinary least squares (OLS) analysis on meteorological yield after the normalization of dependent variables. The relationships between principal components $F_1$–$F_5$ and normalized meteorological yield are shown in Figure 5.

Figure 5 demonstrates that with the increase of the independent variables $F_1$ and $F_3$, the value of $ZY_c$ increases continuously; $F_2$, $F_4$, and $F_5$ with the opposite trend of change, indicating that $ZY_c$ and $F_2$, $F_4$, and $F_5$ have a negative correlation coefficient, and $ZY_c$ has a positive correlation coefficient with $F_1$ and $F_3$.

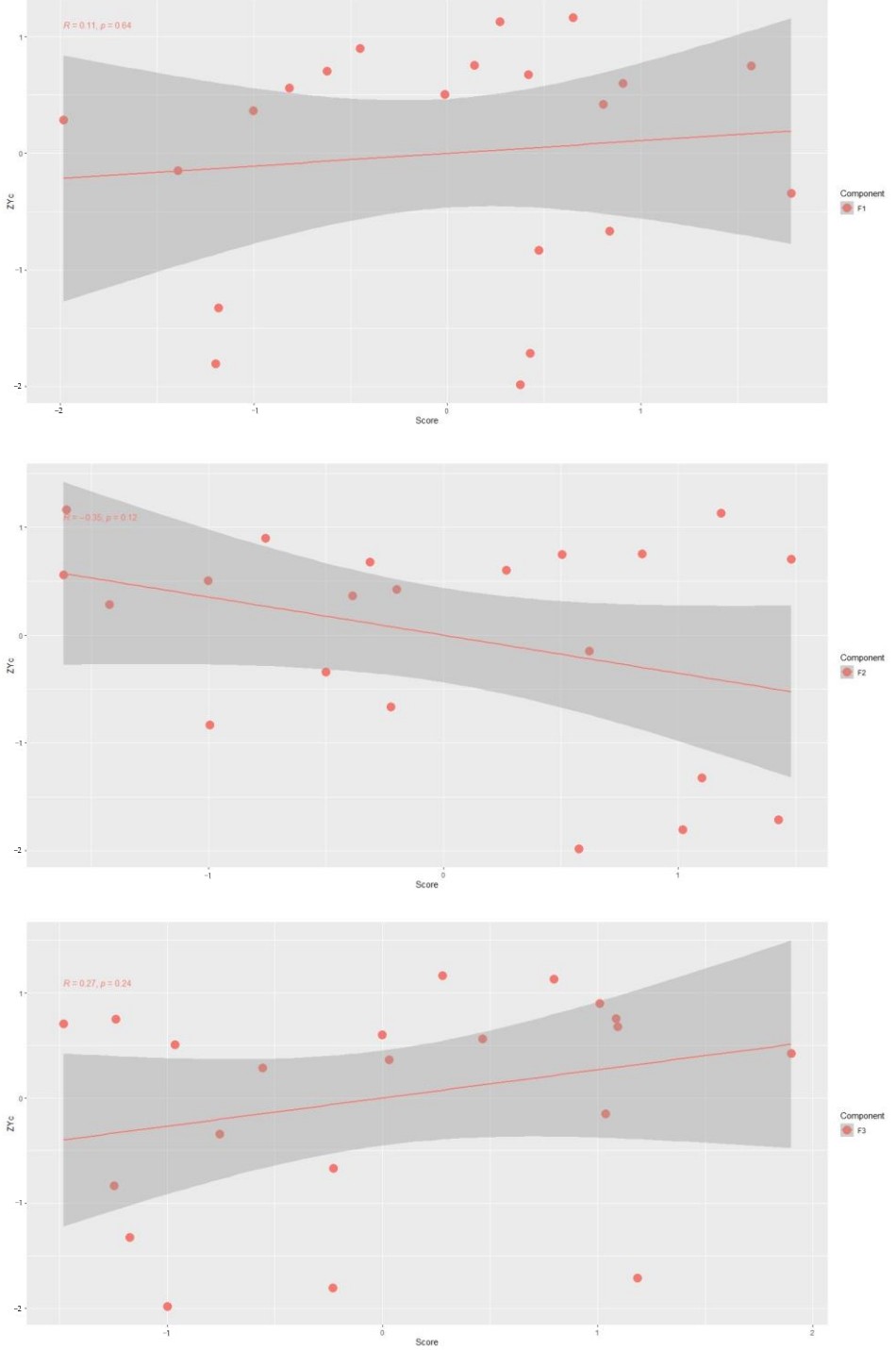

**Figure 5.** *Cont.*

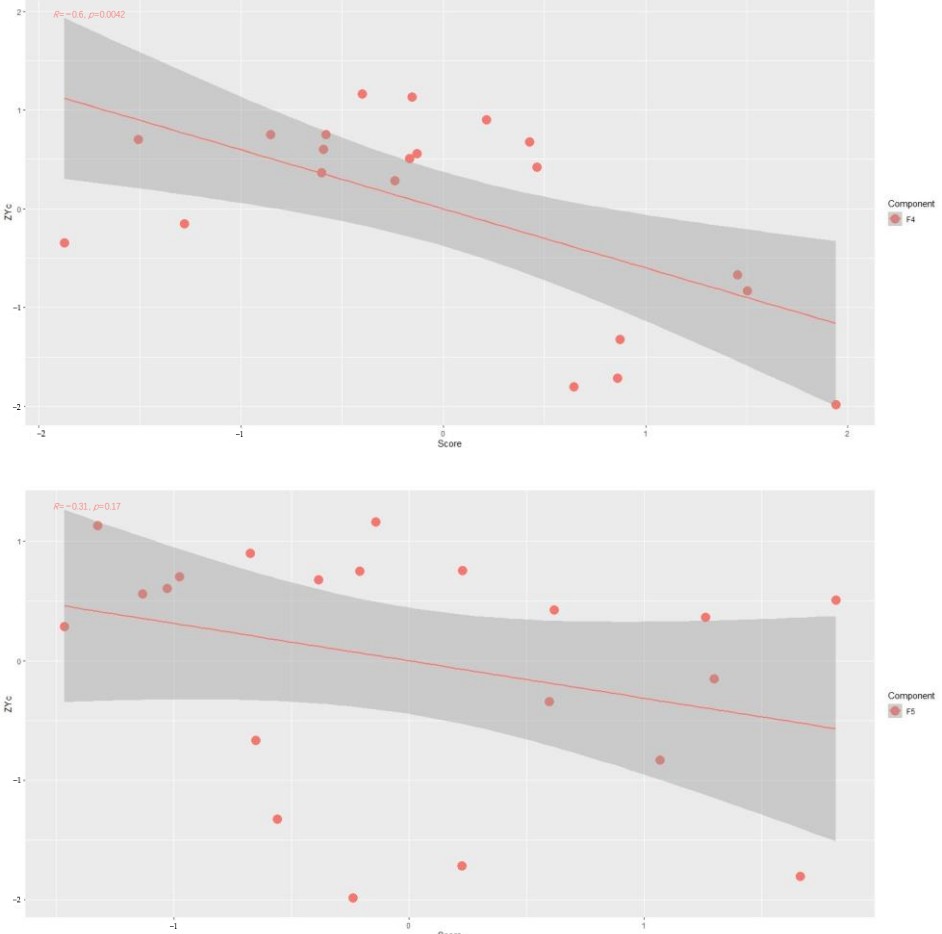

**Figure 5.** Relationships between normalized meteorological yield and principal components $F_1$–$F_5$.

According to the standardized residual analysis (Figure 6), the standardized residual follows the mean value of $-1.73 \times 10^{-16}$ and the normal distribution with a standard deviation of 0.894, which can be approximated following a normal distribution with a mean of zero and a standard deviation of one. The residual distribution satisfies the applicable range of multiple linear regression. Meanwhile, the value of the Durbin–Watson statistic is closer to two and there is less correlation between the residual terms. The value of the Durbin–Watson statistic of the model constructed in this paper is 1.658, which indicates that the residual terms are not correlated.

According to the above analysis, the relationship between the dependent variable $ZY_c$ and the independent variables $F_1$–$F_5$ is linear. The independent variables $F_1$–$F_5$ are not random. Moreover, there is no exact linear relationship between two or more independent variables due to the extraction of the PCA. The expected value of the residual in terms of the independent variable is 0: $E(\varepsilon|F_1, F_2, F_3, F_4, F_5) = 0$. The variance of the residual term is the same for all observations: $E\left(\varepsilon_i^2\right) = \sigma^2$. The residual term is not correlated between the observed values: $E\left(\varepsilon_i\varepsilon_j\right) = 0$, $j \neq i$. The residual term is normally distributed. All the basic assumptions of the multiple linear regression model are consistent. Therefore, the MLR–PCA was established and the backward stepwise regression analysis results are shown in Table 6.

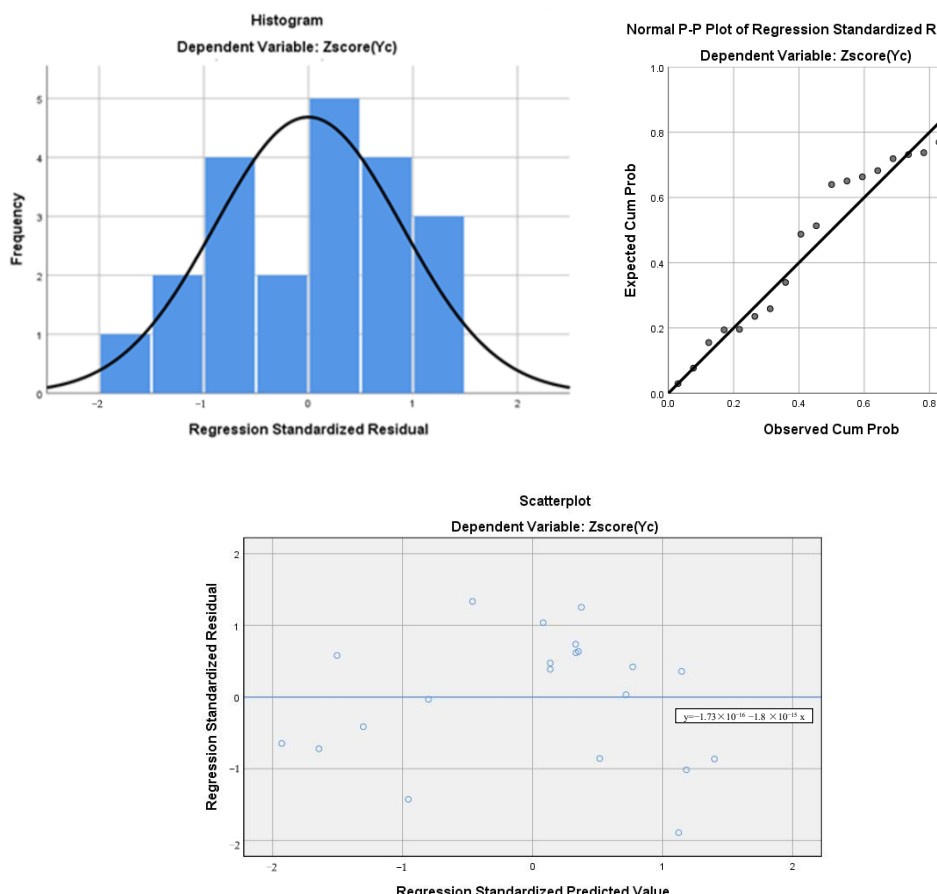

**Figure 6.** The standardized residual analysis.

**Table 6.** Multiple linear backward stepwise regression analysis based on a principal component analysis (B and Beta represent the regression coefficient; Std. represents standard error; Tol. represents tolerance; Sig. represents significance. Besides, *: $p < 0.05$ **: $p < 0.01$ ***: $p < 0.001$).

| Regression Analysis | Unstandardized Coefficient | | Standardized Coefficient | T | Sig. | Covariance Statistics | |
|---|---|---|---|---|---|---|---|
| | B | Std. | Beta | | | Tol. | VIF |
| Constant | $-2.56 \times 10^{-17}$ | 0.144 | | 0.000 | 1.000 | | |
| $F_1$ | 0.108 | 0.150 | 0.108 | 0.722 | 0.481 | 1 | 1 |
| $F_2$ | −0.354 | 0.148 | −0.354 | −2.396 | 0.029 * | 1 | 1 |
| $F_3$ | 0.268 | 0.148 | 0.268 | 1.817 | 0.088 | 1 | 1 |
| $F_4$ | −0.597 | 0.148 | −0.597 | −4.048 | 0.001 *** | 1 | 1 |
| $F_5$ | −0.313 | 0.148 | −0.313 | −2.118 | 0.05 ** | 1 | 1 |

Table 6 shows that the $VIF$ values of meteorological yield after the normalization of the principal components $F_1$–$F_5$ and the dependent variable are all less than 10, indicating that the principal component analysis solved the interference of multicollinearity on the prediction model. The principal components $F_2$, $F_4$, and $F_5$ are negatively correlated with the meteorological yield, which means the higher the growing season heat factor, precipitation factor, and cooling factor are, the lower the meteorological yield is. The principal components $F_1$ and $F_3$ are positively correlated with meteorological yield, which means that the higher temperature factor and humidity factor are, the higher the meteorological

yield is. The regression coefficient is consistent with the results observed in Figure 5. The expression of multiple linear regression is:

$$ZY_c = 0.108F_1 - 0.354F_2 + 0.268F_3 - 0.597F_4 - 0.313F_5 \qquad (19)$$

According to Equations (18) and (19), the standardized linear equation can be expressed as:

$$ZY_c = -0.068ZAAT - 0.075ZAR - 0.236ZMATTT - 0.005ZLAMT - 0.398ZMAP - 0.099ZFFP - 0.619ZATA10 + 0.062ZATMJ + 0.088ZAMT - 0.202ZMATSE - 0.454ZMTPSE + 0.146ZRHJ \qquad (20)$$

According to Equation (20), the coefficients of the accumulated temperature above 10 °C, the June–August precipitation, the annual average precipitation, the June–August mean temperature, and the March–October mean temperature were large. Combined with the conclusion in Section 3.2 "Screening of Key Meteorological Factors", the key meteorological factors were the accumulated temperatures above 10 °C, the March–October mean temperature, and the June–August mean temperature.

The standardized linear equation was destandardized and the expression of the linear equation was obtained as:

$$Y_c = -0.541AAT - 0.172AR - 2.019MATTT - 0.015LAMT - 0.012MAP - 0.034FFP - 0.015ATA10 + 0.126ATMJ + 0.180AMT - 1.440MATSE - 0.048MTPSE + 0.145RHJ + 160.088 \qquad (21)$$

Using Equation (19), in terms of the temperature factor, the whole period of the increase of the temperature factor on apple meteorological yield does not have a significantly positive effect. This result proves that there is not a simple linear relationship between the temperature factor and the meteorological yield in Yantai, but there may be a complex nonlinear interaction. In terms of the growing season heat factor, there is a significantly negative correlation ($p < 0.05$) between the growing season heat factor and apple's meteorological yield. This result proves that the heat supply in the growing period of Yantai may exceed the heat required for the development of apples. High temperatures can easily cause thermal damage, resulting in cell dehydration, affecting the physiological metabolic activities of plants and reducing the yield. In terms of the humidity factor, there is a positive correlation ($p < 0.1$) between the humidity factor and apple's meteorological yield. This shows that Yantai summer humidity may not meet the needs of apple growth. In summer, when the humidity is low and the temperature is high, the stomata on the leaves of plants will close to keep the water in the plant. The closure of stomata will make the leaves unable to capture carbon dioxide, leading to starvation. In terms of precipitation factor, the precipitation factor on apple meteorological yield over the whole period has an extremely significantly negative effect ($p < 0.001$). This means that the precipitation in Yantai exceeds the amount needed for apple growth. Excessive precipitation may cause waterlogging, and the surge in soil water content may cause anoxia, which will cause a series of hazards. In terms of cooling factor, the cooling factor on apple meteorological yield over the whole period has an extremely significantly negative effect ($p < 0.01$). The result indicates that low value of cooling, especially in winter, significantly reduces apple's yield. When apple trees cannot meet the cooling requirements for releasing dormancy, forcibly breaking dormancy will greatly reduce their flowering and fruit setting rates and affect the apple yield.

### 3.4. Establishment and Evaluation of Comparative Models for Forecasting Meteorological Yield

Lasso regression avoids the distortion of the regression results caused by multicollinearity and endogeneity between variables in the regression process by rapidly compressing the coefficients of nonimportant explanatory variables to zero. We chose to compare the fitting effect and prediction results of lasso regression and the MLR–PCA model constructed in this paper and evaluate whether the prediction accuracy was improved by the MLR–PCA model. Lasso regression was conducted with R version 4.2.2 using the packages glmnet for visualization of data (Figure 7).

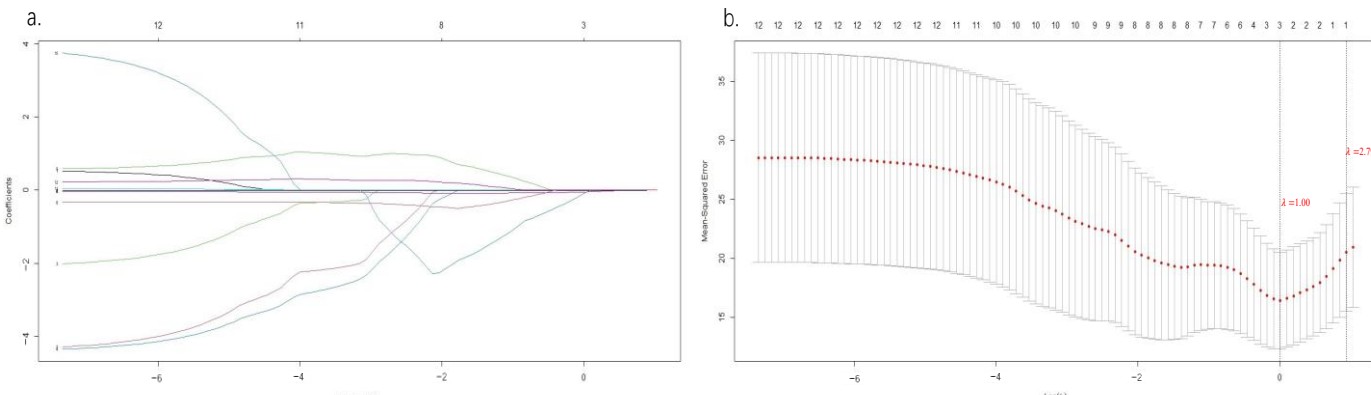

**Figure 7.** Lasso regression adjustment parameter process: (**a**) coefficient plot (**b**) parameter plot.

The lambda for which we choose to cross-validate the model with the smallest mean error is denoted as lambda.min. According to Figure 7b, the value of lambda.min is 1.00, and the corresponding number of important explanatory variables is 3. Combining with Figure 7a, the important explanatory variables were the frost-free period, the accumulated temperature above 10 °C, and the June–August mean temperature. In order to compare the fitting effect and significance of the model [58], the multiple linear regression model was established with important explanatory variables as independent variables and meteorological yield as dependent variables.

The goodness-of-fit and significance of the two models were compared, including MLR–PCA and lasso regression. The results are shown in Table 7. It can be seen from Table 7 that the regression effect of the MLR–PCA model and the lasso regression model is extremely significant ($p < 0.01$). In addition, the goodness-of-fit of MLR–PCA is higher than that of the lasso regression model.

**Table 7.** Comparison of fitting degree and significance test of different models (**: $p < 0.01$).

| Model | $R^2$ | Sig. |
|---|---|---|
| MLR–PCA | 0.663 | 0.003 ** |
| Lasso Regression | 0.534 | 0.004 ** |

In order to verify whether MLR–PCA analysis can effectively improve the model accuracy, the relevant data from Yantai City in 2020 were used [40], the meteorological yield forecasts of apple from the MLR–PCA model and the lasso regression model were calculated and combined with the harmonic weight trend yield forecast method [48–50]. The predicted value of apple yield was obtained, and the yield prediction effects of the three models are shown in Table 8.

By comparing the methods, the predicted value of the MLR–PCA model was 47.256 t·ha$^{-1}$, compared with the actual yield of 44.128 t·ha$^{-1}$. The relative error was 7.089%, which was the smallest among the selected models, indicating that the MLR–PCA model is superior and accurate in forecasting.

**Table 8.** Effect of yield prediction.

| Model | Actual Apple Yield/(t·ha$^{-1}$) | Forecasted Apple Yield/(t ha$^{-1}$) | Trend Yield Forecast/(t·ha$^{-1}$) | Meteorological Yield Forecast/(t·ha$^{-1}$) | Relative Error |
|---|---|---|---|---|---|
| MLR–PCA | 44.128 | 47.256 | 58.481 | −11.225 | 7.089% |
| Lasso Regression | 44.128 | 48.116 | 58.481 | −10.365 | 9.038% |

## 4. Conclusions

In this work, the influence of different meteorological factors on the apple yield in Yantai was investigated numerically and experimentally, and the weights of the influence of different meteorological factors were analyzed. The MLR–PCA model and prediction results were validated by yearbook data. The main scientific and valuable conclusions are as follows:

(1) Apple yield is in a stable growth stage. The trend yield is the decisive factor affecting apple yield, and the increase of trend yield drives the increase of apple yield. Meteorological output has been decreasing annually since 2014, which inhibits the growth of apple yield.

(2) An accumulated temperature above 10 °C has extremely significant inhibitory effects on the meteorological yield of apples ($p < 0.01$) and the frost-free period has a significant inhibitory effect on the meteorological yield of apples ($p < 0.05$). There are certain correlations among different meteorological factors. Among them, the accumulated temperature above 10 °C, the March–October mean temperature, and the June–August mean temperature are the key meteorological factors affecting the meteorological yield of apples.

(3) The MLR–PCA method for dimension reduction includes 12 indicators from five separate comprehensive meteorological factor indicators which are: temperature factor, growing season heat factor, humidity factor, precipitation factor, and cooling factor. The precipitation factor regression coefficient has the highest absolute value, which proves that precipitation factor is the key indicator that most affects meteorological yield. By parity of reasoning, in the growing season, heat factor is the second, cooling factor is the third, humidity factor is the fourth, and temperature factor is the lowest key indicator affecting meteorological yield. The results demonstrate that precipitation factor, growing season heat factor, and cooling factor are beyond the optimal adaptation range of apples, which results in the decrease of apple yield. For each 1% increase of the three factors apple yield is reduced by 0.597%, 0.354%, and 0.313%, respectively. However, humidity factor and temperature factor do not meet the demand of apples, and each 1% increase of the humidity factor and temperature factor increases apple yield by 0.268% and 0.108%, respectively.

(4) The MLR–PCA and lasso regression methods were used to make predictions and comparisons. The apple yield predicted by the MLR–PCA model was 47.256 t·ha$^{-1}$, compared with the actual yield of 44.128 t·ha$^{-1}$. The relative error was only 7.089%. The prediction effect was better than other models. This work provides theoretical support for the future prediction of apple yields.

**Author Contributions:** Conceptualization, X.H.; methodology, X.H. and L.C.; validation, X.H. and L.C.; formal analysis, X.H. and N.W.; data curation, X.H. and C.W.; writing—original draft preparation, X.H.; supervision, W.K. and C.W. All authors have read and agreed to the published version of the manuscript.

**Funding:** This research was funded by the National Innovation Training Program for College Students (202110019088), the Yantai School-Land Integrated Development Project (2019XDRHXMPT30), and the URP Project of China Agricultural University (U2021057).

**Institutional Review Board Statement:** Not applicable.

**Informed Consent Statement:** Not applicable.

**Data Availability Statement:** The datasets generated and analyzed in the current study are publicly available and are also available from the corresponding author upon reasonable request.

**Acknowledgments:** The authors would like to thank the Yantai Bureau of Statistics for collecting the apple production dataset. The authors are grateful for the comments from anonymous reviewers and the editors.

**Conflicts of Interest:** The authors declare no conflict of interest.

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
