# Peer review of "Effects of Meteorological Factors on Apple Yield Based on Multilinear Regression Analysis: A Case Study of Yantai Area, China"

_atmosphere, doi:10.3390/atmos14010183_

Round 1

Reviewer 2 Report

Dear authors:

Your article is very interesting, and I am grateful for the opportunity to read it. I think that the research subject is interesting, and the results of the research give a lot of new information and possibilities for further analysis.

This paper mainly used the principal component regression method to analyze the influence of different meteorological factors on Yantai apple yield, including temperature, precipitation, and humidity. There are several obvious problems:

1.  From my point of view, the corresponding author is marked with * rather than 1 on line 5.

2.  The yield separation model was innovatively proposed and used in this paper; it was illustrated to some extent. However, as the highlight of the whole paper, the presentation of relevant mathematical models and formulas is lacking. Similarly, when comparing the prediction accuracy of different models in the results and analysis part, the harmonic weight trend yield forecast method was not mentioned, so it is suggested to supplement and explain in the method introductionpart.

3.  In the Result and analysis part of this paper, the description of some figures or tables only gives a simple description of the results without in-depth analysis. For example, "Table 3 shows that the ??? values of different meteorological factors are greater than 10, and the ??? values of some meteorological factors are far more than 100, indicating that there is serious multicollinearity among different meteorological factors in Yantai." Why is there such a severe collinearity problem in Yantai meteorological factors? What is the cause of this phenomenon?

4.  In the Discussion and Conclusion part of this paper, the discussion contents are repeated. For example, "The apple yield model established in this study can accurately predict the short-term apple yield in the future. To improve the accuracy of the model, more factors need to be considered" and "In this work, the utilization of meteorological resources is evaluated from the perspectives of the heat factor, humidity factor and high-temperature factor. In the future, meteorological resources need to be evaluated from more perspectives". The author considers including more meteorological factors for analysis and assessment, so it is suggested to combine the two paragraphs into one.

In detail:

In figures, the axis title does not agree with the content of the article. For example, the vertical coordinate of Figure 3 should indicate whether it is meteorological yield, trend yield, or actual yield mentioned in the article. Meanwhile, this figure in the horizontal coordinate does not use the subscript as in the article. In addition, the information shown in the figures is inconsistent with the conclusion. For example, the location circled in red in Figure 1 is not the location described in the picture title, and Figure 2 has a similar problem.

In tables, the names in the table contents are inconsistent. For example, Table 7 and Table 8 have different names for the second row of the "model" column, which is called "Lasso regression" in Table 7, while in Table 8, it is called "Stepwise regression." Besides, please note the uniformity of formatting in the table. For example, in Table 2, the variable names in the first row are bold. In Table 3, VIF is available in italic and non-italic formats.

- Section 1

-The content needs to be analyzed using data from the same period as the full text. Such as, "Provincial-level meteorological and yield data from 2007 to 2019 were republished by Yantai Bureau of Statistics" and "Meanwhile, China is the largest apple producer in the world. Particularly, the northern regions contribute more than 70 percent of the apple yield to China".

-On line 52, instead of "scholars at home and abroad" use "scholars in different countries" or "international scholars" etc.

-Section 2

  -In the formula (3) and (9), all of which use the capital letter Xi, which is easy to be confused. In addition, in Section 3, the symbol representing the meteorological factor is also Xi, which is easily confused with  Xi here. It is recommended to assign different symbols to variables with different meanings.

- Section 3

  -Section 3.1-

-Figure 2 contains obvious errors. Besides, in the interpretation of the orange line on line 211, the unit of 0 t·ha-1 is misrepresented. Please review and correct them.

-Even if the orange line in Figure 2 is accurately moved to the corresponding position, it is difficult to identify the value of apple meteorological yield between 2008 and 2010. Is there a better way to represent it?

-Section 3.2-

-Please explain the reason why meteorological factors with a correlation coefficient greater than 0.68 are defined as the key meteorological factors. Is it supported by other literature or the author's subjective judgment? It is not clear. Can this criterion be applied to other models?

-Section 3.3-

- Please keep the pictures in the same size in Figure 3.

Summarizing

I find your article very good. I really like your article and appreciate your work. It is an interesting topic, and the conclusions could open the way for further research. You have to make some changes, especially in Results and analysis, Discussion and Conclusion. My general opinion and my assessment of your research and the whole article are more than positive.

Reviewer 3 Report

The authors have meticulously carried out many statistical analyses in the article Effects of Meteorological Factors on Apple Yield Based on Principal Component Regression Analysis: A Case Study of Yantai Area, China. Therefore, I found the paper to be of value and to contain a really complex analytical study of the meteorological factors affecting apple production in a particular region in China. However, despite this, I have some major comments for the authors:

1)      The research hypothesis is missing when defining the purpose of the study. Therefore, I recommend the authors to include a statistically testable hypothesis in the introduction chapter. The authors do much analysis in this paper, and I think they should clearly state their hypothesis at the beginning of the research.

2)      The map in figure 1 looks unprofessional. Please use GIS systems to create a map with the study region correctly.

3)      In lines 199-205, you give the meteorological factors you used for your analyses. Although the units of each factor are fairly self-explanatory, for scientific correctness, you should give the unit of each meteorological factor here.

4)      In figure 2 you show the Actual yield. However, there is no information in the manuscript where these values were derived from? Please provide a relevant source or method of calculation.

5)      I ask the authors to separate the discussion section from the conclusions. Furthermore, the article, as it stands, lacks a decent discussion. The authors make almost no reference to previously published literature. How do your results compare with previously published results? Please significantly restructure the discussion section, as it does not meet the standards of a research article in its current form.

Round 2

Reviewer 1 Report

The authors implemented all the suggestions and highly improved clarity of the manuscript.

Reviewer 2 Report

The problems existing in the previous manuscript have been corrected. The content and logic of the revised manuscript are innovative. Therefore, I advocate acceptance.

Reviewer 3 Report

I appreciate the authors' contribution to improving the manuscript Effects of Meteorological Factors on Apple Yield Based on Multilinear Regression Analysis: A Case Study of Yantai Area China. However, I am not fully satisfied with the authors' responses to my comments. I think they have largely improved the paper; however, I still have a few observations:

I still think the authors could have worked on better describing the research aim and including the research hypothesis in the introduction. However, this part of the text has not actually been changed in the new version of the manuscript.

Still, the discussion should be developed, and the authors should relate their results more to the world literature. In my opinion, the whole manuscript is of a high level - especially the very thorough statistical analyses; however, a decent discussion of the results is still missing in this article.

I also ask the authors to review the entire publication text carefully about formatting. For example, there must not be a situation like in the abstract where there is the number 10 on one line and only on the next line the unit, i.e. ℃.
